# Association between depression and quality of life in stroke patients: The Korea National Health and Nutrition Examination Survey (KNHANES) IV–VII (2008–2018)

**Sun Woo Kang[1], Sook-Hyun Lee[2], Ye-Seul Lee[2], Seungwon Kwon[3], Peggy Bosch[4], Yoon Jae Lee[2], In-Hyuk Ha[2]***

**1** Jaseng Hospital of Korean Medicine, Seoul, Republic of Korea, **2** Jaseng Spine and Joint Research Institute, Jaseng Medical Foundation, Seoul, Republic of Korea, **3** Department of Cardiology and Neurology, Kyung Hee University Korean Medicine Hospital, Seoul, Republic of Korea, **4** Donders Institute for Brain, Cognition and Behaviour, Radboud University, Nijmegen, Nijmegen, The Netherlands

* hanihata@gmail.com

## Abstract

### Background

Stroke and depression are common diseases that affect quality of life (QoL). Although some recent studies have investigated the association between the two diseases, studies that examined the association between stroke, depression, and QoL are rare, with large-scale national-level studies lacking. We aimed to investigate the association between depression and QoL in stroke patients.

### Methods

Data from the Korea National Health and Nutrition Examination Survey (KNHANES) IV–VII conducted in 2008–2018 were used, and 45,741 adults who were aged >40 years and had no missing data for stroke and depression were included in the analysis. The participants were first grouped by prevalence of stroke, and further divided by prevalence of depression.

### Results

The overall prevalence of stroke was 3.2%, and the incidence was 9% higher in men than in women. Multiple logistic regression was performed after adjusting for demographic factors, health-related factors, and disease-related factors. The results confirmed that the stroke group with depression had a lower overall health-related quality of life, measured using EQ-5D, score compared to the stroke group without depression (-0.15). Moreover, the concurrent stroke and depression treatment group had the highest OR of 7.28 (95% CI 3.28–16.2) for the anxiety/depression domain.

**Data Availability Statement:** The third party data underlying this study are available from the Korea Disease Control and Prevention Agency (KDCA).

Researchers can download the data from the website (https://knhanes.kdca.go.kr/knhanes/eng/index.do) once they present their identification details, e.g., name and email address on the website. Inquiries on the database are handled by the Division of Health and Nutrition Survey and Analysis, KDCA. The authors confirm that they had no special access privileges which others would not have.

**Funding:** This research was funded by the Jaseng Medical Foundation, Republic of Korea. (JS-RP-2021-24). The funders had no role in study design, data collection and analysis, decision to publish, or preparation of the manuscript.

**Competing interests:** The authors have declared that no competing interests exist.

## Conclusion

Depression was strongly associated with QoL in stroke patients. This association was more evident in stroke patients undergoing treatment for depression. Thus, clinical approaches that take QoL into consideration are needed for stroke patients with depression.

## Introduction

Stroke refers to an abrupt onset of local neurological defect caused by abnormal cerebral blood flow. The burden of stroke is high not only in Korea but worldwide, and approximately 105,000 people in Korea are newly diagnosed or have a recurrence of stroke in Korea every year [1]. Despite such grave socioeconomic burden, stroke has rarely been studied in Korea. Moreover, stroke is the second leading cause of death in the country and a major culprit of physical disability. The consequent physical disabilities deteriorate the quality of life (QoL) of the patient as well as their family. For these reasons, understanding the QoL of people suffering from sequelae of stroke is crucial to planning public health services and assessing disease management [2].

Although stroke mortality is on a decline, there are regional differences in the rates, and still 30 patients per 100,000 population die from stroke [2]. In addition, Hong reported the estimated incidence to be 216 per 100,000 person-years. The incidence rate per 100,000 person-years dramatically increased with age, from 20 cases among those aged ≤44 years to 3,297 among those aged ≥85 years [1]. Due to the high incidence, the total economic burden of stroke, which includes medical, non-medical, and indirect cost, was estimated to be 4.2 billion US dollars in 2008 [3]. In a study on the recurrence of stroke in Asia, Chin reported a rate reaching 25.4% in some regions, with a two-year recurrence rate of 12.9% and five-year recurrence rate of 16% [4]. Despite the declining mortality rate, the number of people suffering from sequelae after stroke is growing due to the high incidence rate and recurrence rate.

Depression is characterized by a sad mood and physical inactivity. It is a common mental disorder affecting approximately 350 million people worldwide. Depression was pinpointed as the leading cause of disability worldwide by the World Health Organization (WHO) in 2015, with a prevalence of 7.5% among all individuals with a disability [5]. In the 2020 study by Kim, the prevalence of depression in a one million sample population rose from 2.8% in 2002 to 5.3% in 2013, and the prevalence increased with advancing age and was generally higher in women than in men in most age groups [6]. Furthermore, depression is not only often a primary disorder but also a secondary disorder (caused by other diseases).

One of the consequences of stroke such as the newly acquired physical disabilities and subsequent social isolation caused by stroke, trigger psychological responses such as anger and despair; in addition, physical inactivity and loss of physical sensations due to physical disabilities also cause depression. These findings imply the substantial impairment of QoL by post-stroke depression. Moreover, post-stroke depression aggravates the burden on caregivers [7].

As a result of the rising prevalence of stroke worldwide, the prevalence of post-stroke depression is also increasing, and many studies have examined this condition [8–10]. In Korea, the prevalence of depression was higher among patients diagnosed with stroke, which shows that stroke is an important risk factor for depression. Studies that investigated the effects of stroke and depression on QoL independently have reported that both diseases indeed reduce QoL [11]. Hence, it is hypothesized that QoL would differ between stroke patients with and without depression. One study reported that a steady management of depression can have

a tremendous impact on facilitating functional recovery and improving the QoL of community-dwelling stroke patients [12]. However, not many studies have examined whether QoL differs according to depression in stroke patients. Moreover, studies investigating the differences in QoL according to stroke and depression are rare, and even the existing studies failed to present consistent results, with national-level studies virtually lacking. Therefore, this study aimed to analyze the association between depression and QoL in stroke patients using a nationally representative adult population.

## Materials and methods

### Database

The generation of the database KNHANES IV–VII was conducted by the KDCA, and a written consent was obtained from all participants. All survey protocols were approved by the institutional review board (IRB) of the KCDC (approval numbers: 2008-04EXP-01-C, 2009-01CON-03-2C, 2010-02CON-21-C, 2011-02CON-06-C, 2012-01EXP-01-2C, 2013-07CON-03-4C, 2013-12EXP-03-5C, and 2018-01-03-P-A). Each participant voluntarily participated and provided a written informed consent before participating in the study. This study was given a formal waiver from the Institutional Review Board of Jaseng Hospital of Korean Medicine in Seoul, South Korea (JASENG 2021-05-017).

### Study participants

Data from the KNHANES IV–VII, conducted from January 2008 to December 2018, was used in this study. KNHANES database is built based on a complex sample survey for which the national population is sampled using three-stage cluster stratification method. Stratification variables for building this database included principal administrative regions and type of residence in the first strata; type of household and household member characteristics in the second strata; and subdivisions in the administrative regions in the third strata [13]. For the analysis in this study, sample weights, variance strata, and stratification variables were applied to obtain representativeness on the national population. In KNHANES IV-VII, in which 93,028 participants across all ages were surveyed, we extracted the data of 45,741 adults aged 40 years or older who participated in the health examination. Of 93,028 participants, 42,619 participants aged <40 years and 4,668 participants with missing data on stroke and depression were excluded. (Fig 1). From 45,741 adults included in this study, the participants were first grouped by prevalence of stroke, and further subdivided by prevalence of depression.

### Assessment of covariates

The participants' demographic factors and health-related factors were assessed using health interviews and examinations. Demographic factors included age, sex, area of residence, marital status, household income, employment status, and education level. The participants were classified as male or female, and age was categorized into 40–49 years, 50–59 years, 60–69 years, 70–79 years, and $\geq$ 80 years. Area of residence was classified based on the administrative districts into urban (dong) and rural (eup/myeon) [14, 15]. Household income level was divided into quartiles of equalized household income (low, mid-low-, mid-high, and high) according to the KNHANES. Education level was divided into elementary school or lower, middle school, high school, and college or higher. Employment status was categorized into employed and unemployed as used in the KNHANES. Marital status was divided into three groups: married-cohabit for those who are currently married; married-no cohabit for those who were

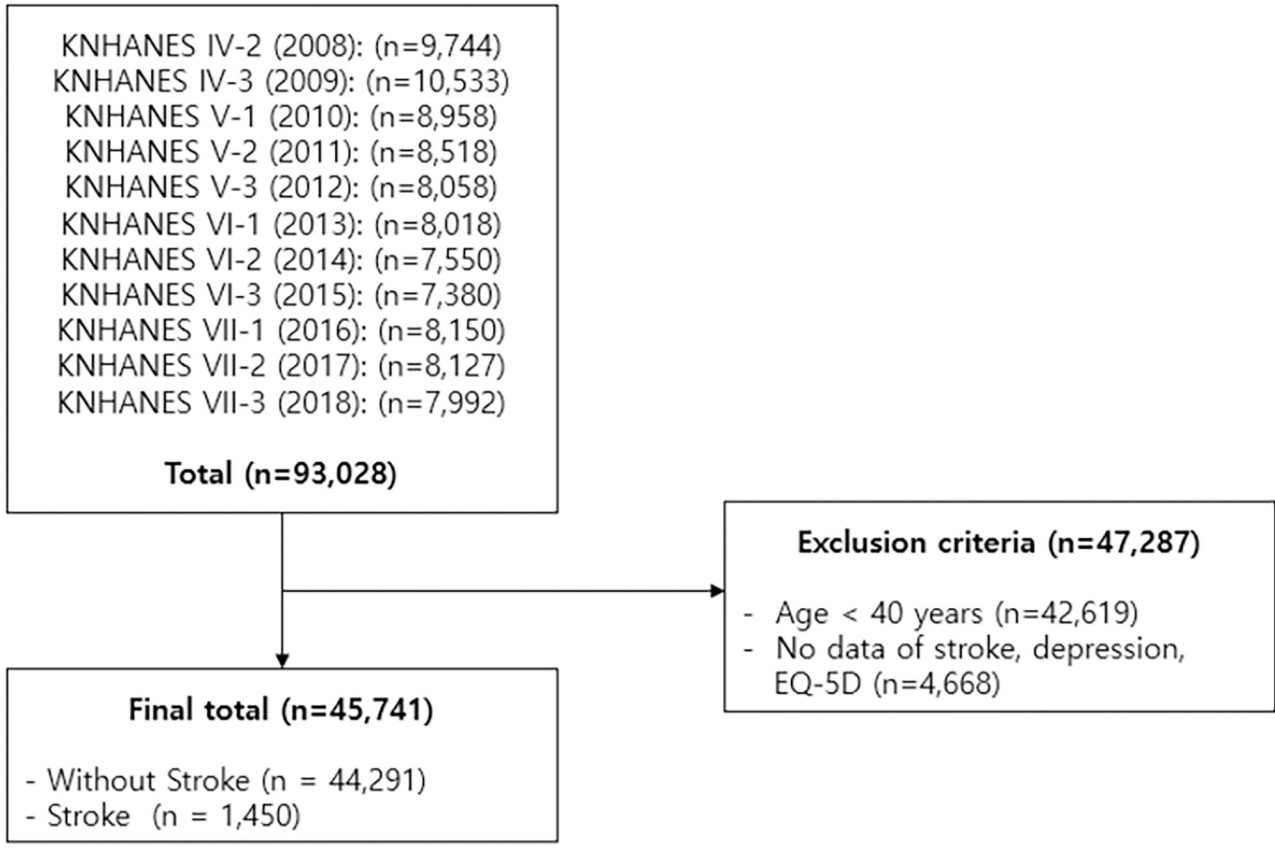

**Fig 1. Flow diagram showing the process of participant selection and exclusion.**

married in the past but are currently separated, widowed, or divorced; and unmarried for those who have never been married.

Health-related factors included BMI, alcohol consumption, smoking, physical activity (walking), muscle training, stress level, and depression. BMI was calculated by dividing weight (kg) by height (m) squared using the available anthropometric measurements, and based on the WHO criteria, it was classified into underweight ($<18.5$ kg/m$^2$), normal (18.5 to $<25$ kg/m$^2$), and overweight and obese ($\geq25$ kg/m$^2$) [31]. Alcohol consumption was defined as the intake of alcohol more than once a month on average for the past year, and smoking status was classified into non-smoker, ex-smoker, and current smoker. Physical activity (walking), muscle training, and depression were categorized using yes/no answers, and stress was divided into high or low.

## Definition of stroke and depression

KNHANES survey items regarding the patients' diagnosis history of stroke and depression were adopted to define the diseases in this study. Since this survey was conducted on each participant who were answering from the patient's point of view, each item was answered within the range of yes or no, and no specific details were included for each diagnosis. The survey item on the medical diagnosis of the disease within the two-weeks period before answering the survey was used to define diagnosis of stroke and/or depression of the patient. The survey item

on the history of medical treatments for stroke and/or depression was used to define treatment history of stroke and/or depression.

## Assessment of QoL

Health-related QoL (HRQoL) was assessed using the EQ-5D, which was included as part of the KNHANES survey. Since this tool was officially included in KNHANES survey, it was possible to compare HRQoL across different patient groups within the participants. The EQ-5D evaluates HRQoL based on mobility, self-care, usual activities, pain/discomfort, and anxiety/depression, and it is a widely employed tool for assessing HRQoL across different disease states, allowing for comparisons across patient groups and diseases. It was developed to measure overall health, and it is generally used to measure HRQoL of patients with chronic conditions. In this study, we used the weighted values generated by the Korea Disease Control and Prevention Agency (KDCA) in the KNHANES. Golicki validated the EQ-5D for measuring HRQoL of stroke patients [16]. In addition, Franklin also validated the instrument for measuring HRQoL in patients with depression [17]. In this study, we used the EQ-5D to measure QoL in relation to depression in stroke patients.

## Statistical analysis

The KNHANES uses a complex sample design, so statistical analyses for complex sample designs were performed in consideration of stratification and cluster variables. The differences in the participants' characteristics according to stroke or depression were analyzed with the Rao-Scott Chi-Square test or t-test. Continuous variables are presented as mean and standard deviation, and categorical variables are presented as number and percentage (N, %). Moreover, the association between stroke and depression was analyzed using a logistic regression with EQ-5D score as the dependent variable and stroke and depression as the major independent variables and after adjusting for demographic factors, health-related factors, and disease-related factors. The results are presented as odds ratio (OR) and 95% confidence interval (CI) to assess the association between stroke and depression and relevant factors. We performed multiple linear regression for the total EQ-5D index score to calculate the regression coefficient and 95% CI.

All statistical analyses were performed using the SAS V9.4 (SAS Institute Inc, Cary, NC, USA) software, and significance was set at below 0.05.

## Results

### Demographic characteristics and QoL of stroke patients

Of all participants, 3.2% had stroke, and there were 9% more men than women. The mean age was 56.0±11.3 years in the non-stroke group and 66.3±10.2 years in the stroke group (p<0.0001). In the stroke group, the most common income level was "low" (44.7%), and the most common education level was "elementary school or lower" (52.2%). The majority of the participants in the stroke group were unemployed (67.2%) (p<0.0001).

Regarding health-related factors, the mean BMI value was 24.5±3.3 in the stroke group, and 47.7% of those in the stroke group did not drink alcohol. There were more non-smokers (47.9%) than ex-smokers or current smokers in the stroke group (p<0.0001).

Regarding the EQ-5D, the proportions of participants in the stroke group who had difficulties in each of the five dimensions (mobility, self-care, usual activity, pain/discomfort, anxiety/depression) were markedly higher at 49.4%, 25.5%, 39.7%, 49.8%, and 24.2%, respectively, than those in the non-stroke group. The total EQ-5D score was lower in the stroke group (0.81

± 0.21) than in the non-stroke group (0.94±0.12), showing that stroke patients have a poorer QoL.

In 2008, there were 161 stroke patients, accounting for 8.6% of the participants. In 2018, there were 145 stroke patients (11.5%). This implies that there was a general upward trend over a period of 11 years (Table 1).

## Demographic characteristics and QoL of stroke patients with or without depression

The stroke without depression group comprised 57.3% men and 42.7% women, while the stroke with depression group comprised 24.2% men and 75.8% women, showing a higher percentage of women with depression. Most of the participants in the two groups were unemployed, at 65.5% and 86.2%, respectively, showing a higher percentage of unemployed individuals in the stroke with depression group (p<0.0001).

In the stroke without depression and stroke with depression groups, the proportions of non-drinkers were 46.5% and 61.0%, respectively, and the proportions of non-smokers were 46.2% and 67.6%, respectively.

Regarding the EQ-5D, the proportions of participants in the stroke without depression group who had difficulties in each of the five dimensions (mobility, self-care, usual activity, pain/discomfort, anxiety/depression) were 48.3%, 24.4%, 38.5%, 48.3%, and 22.1%, respectively. In the stroke with depression group, the proportions of participants who had difficulties in each of the five dimensions were 61.1%, 36.9%, 53.7%, 66.4%, and 48.1%, respectively. The total EQ-5D score was lower in the stroke with depression group (0.7±0.3) than the stroke without depression group (0.8±0.2), showing that stroke patients with depression have a poorer QoL.

In 2008, there were 147 stroke patients without depression (8.7%) and 14 stroke patients with depression (7%), and in 2018, there were 127 stroke patients without depression (11.2%) and 18 (15.2%) stroke patients with depression. This implies that there was a general upward trend over a period of 11 years; particularly, the proportion of stroke patients with depression steadily increased over the years (Table 2).

## Distribution of total EQ-5D score according to depression in stroke patients by year

Fig 2 shows the total EQ-5D scores of the stroke with depression and stroke without depression groups from 2008–2018. From 2008–2010, the score distributions were generally similar, and the total EQ-5D score was generally lower in the stroke with depression group compared to that in the stroke without depression group, which suggests that stroke with depression is more strongly associated with poorer QoL, compared to stroke without depression.

## Logistic regression for the association of QoL with stroke and depression

Table 3 shows the association of each of the five domains of QoL with four groups of Korean adults aged 40 years and over: no stroke-no depression group, stroke without depression group, depression group, and stroke with depression group. In Model 2 adjusted for demographic factors, the OR for the mobility domain of the EQ-5D with reference to the no stroke-no depression group was 3.60 (95% CI 2.22, 5.85) in the stroke with depression group, higher than 2.42 (95% CI 2.06, 2.86) in the stroke without depression group and 2.06 (95% CI 1.81, 2.35) in the depression group. Similarly, the OR for the self-care domain was 6.00 (95% CI 3.75–9.61) in the stroke with depression group, higher than 3.53 (95% CI 2.93–4.26) in the

**Table 1. Characteristics of the study population according to stroke.**

| Variable | Total n (%) | Without Stroke n (%) | Stroke n (%) | p-value |
|---|---|---|---|---|
| **Total** | 45,741 | 44291 (96.8%) | 1450 (3.2%) | |
| **Age**(mean±SD) | 56.2±11.4 | 56.0±11.3 | 66.3±10.2 | <.0001 |
| 40–49 | 12008 | 11949 (35.2%) | 59 (6.7%) | <.0001 |
| 50–59 | 12185 | 11972 (30.8%) | 213 (20.3%) | |
| 60–69 | 11119 | 10636 (18.5%) | 483 (30.3%) | |
| 70–79 | 8277 | 7756 (12.1%) | 521 (31.1%) | |
| ≥80 | 2152 | 1978 (3.5%) | 174 (11.7%) | |
| **Gender** | | | | |
| male | 19526 | 18772 (47.6%) | 754 (54.5%) | <.0001 |
| female | 26215 | 25519 (52.4%) | 696 (45.5%) | |
| **Region** | | | | |
| Urban area | 34569 | 33546 (79.6%) | 1023 (76.1%) | 0.0071 |
| Rural area | 11172 | 10745 (20.4%) | 427 (23.9%) | |
| **Marital status** | | | | |
| Married-cohabit | 36052 | 35043 (80.8%) | 1009 (69.1%) | <.0001 |
| Married-no co habit or bereaved or divorced | 445 | 431 (1.0%) | 14 (0.9%) | |
| Unmarried | 9179 | 8753 (18.2%) | 426 (30.0%) | |
| **Income** | | | | |
| Low | 11394 | 10714 (19.9%) | 680 (44.7%) | <.0001 |
| Lower middle | 11334 | 10970 (24.8%) | 364 (25.1%) | |
| Higher middle | 10847 | 10626 (26.3%) | 221 (16.5%) | |
| High | 11710 | 11541 (28.9%) | 169 (13.7%) | |
| **Employment** | | | | |
| Unemployed | 19369 | 18374 (36.3%) | 995 (67.2%) | <.0001 |
| Employed | 26258 | 25810 (63.7%) | 448 (32.8%) | |
| **Education Level** | | | | |
| Elementary school or less | 15885 | 15067 (27.3%) | 818 (52.2%) | <.0001 |
| Middle school | 6601 | 6356 (14.2%) | 245 (18.2%) | |
| High school | 13238 | 12972 (32.9%) | 266 (20.2%) | |
| College or over | 9881 | 9767 (25.6%) | 114 (9.4%) | |
| **BMI (kg/m2) (Mean±SD)** | 24.1± 3.2 | 24.1± 3.2 | 24.5± 3.3 | <.0001 |
| Underweight <18.5 | 1260 | 1224 (2.6%) | 36 (2.2%) | 0.0010 |
| Normal (18.5–24.9) | 28191 | 27380 (61.6%) | 811 (56.4%) | |
| Overweight and Obese (≥25.0) | 16137 | 15548 (35.8%) | 589 (41.4%) | |
| **Alcohol consumption** | | | | |
| Non-drinker | 15766 | 15038 (29.8%) | 728 (47.7%) | <.0001 |
| ≤1 drink/mo | 11954 | 11676 (26.2%) | 278 (19.4%) | |
| 2 drinks/mo to 3 drinks/wk | 13962 | 13651 (35.1%) | 311 (23.7%) | |
| ≥4 drinks/wk | 3773 | 3652 (8.9%) | 121 (9.2%) | |
| **Smoking** | | | | |
| Nonsmoker | 27142 | 26421 (55.6%) | 721 (47.9%) | <.0001 |
| EX-smoker | 8344 | 7964 (19.5%) | 380 (29.2%) | |
| Current smoker | 9976 | 9643 (24.9%) | 333 (22.9%) | |
| **Physical activity (walking)** | | | | |
| No | 8855 | 8478 (18.7%) | 377 (25.5%) | <.0001 |
| Yes | 36683 | 35622 (81.3%) | 1061 (74.5%) | |
| **Days of strength exercise** | | | | |
| No | 34963 | 33798 (74.7%) | 1165 (79.6%) | 0.0007 |

*(Continued)*

**Table 1.** (Continued)

| Variable | Total<br>n (%) | Without Stroke<br>n (%) | Stroke<br>n (%) | p-value |
|---|---|---|---|---|
| Yes | 10624 | 10348 (25.3%) | 276 (20.4%) | |
| **Stress level** | | | | |
| Low | 34763 | 33681 (75.9%) | 1082 (77.0%) | 0.4468 |
| High | 10699 | 10348 (24.1%) | 351 (23.0%) | |
| **Depression** | | | | |
| No | 43446 | 42125 (95.5%) | 1321 (91.7%) | <.0001 |
| Yes | 2295 | 2166 (4.5%) | 129 (8.3%) | |
| **Heath problems (EQ-5D)** | | | | |
| **Mobility** | | | | |
| No | 35764 | 35080 (83.1%) | 684 (50.6%) | <.0001 |
| Yes | 9977 | 9211 (16.9%) | 766 (49.4%) | |
| **Self-care** | | | | |
| No | 42847 | 41788 (95.5%) | 1059 (74.5%) | <.0001 |
| Yes | 2894 | 2503 (4.5%) | 391 (25.5%) | |
| **Usual activity** | | | | |
| No | 39639 | 38797 (90.1%) | 842 (60.3%) | <.0001 |
| Yes | 6102 | 5494 (9.9%) | 608 (39.7%) | |
| **Pain/discomfort** | | | | |
| No | 32449 | 3174 3(74.7%) | 706 (50.2%) | <.0001 |
| Yes | 13292 | 12548 (25.3%) | 744 (49.8%) | |
| **Anxiety/depression** | | | | |
| No | 39816 | 38734 (88.7%) | 1082 (75.8%) | <.0001 |
| Yes | 5925 | 5557 (11.3%) | 368 (24.2%) | |
| Total EQ-5D score | 0.93± 0.13 | 0.94± 0.12 | 0.81± 0.21 | <.0001 |
| **Cycle (year)** | | | | |
| 2008 | 4499 | 4338 (8.3%) | 161 (8.6%) | 0.0121 |
| 2009 | 4996 | 4850 (8.6%) | 146 (7.5%) | |
| 2010 | 4228 | 4121 (8.7%) | 107 (6.3%) | |
| 2011 | 4252 | 4121 (8.9%) | 131 (8.5%) | |
| 2012 | 4003 | 3910 (8.8%) | 93 (6.2%) | |
| 2013 | 3711 | 3568 (8.8%) | 143 (11.3%) | |
| 2014 | 3567 | 3440 (8.6%) | 127 (8.6%) | |
| 2015 | 3758 | 3629 (9.1%) | 129 (10.0%) | |
| 2016 | 4109 | 3983 (9.9%) | 126 (10.3%) | |
| 2017 | 4270 | 4128 (10.0%) | 142 (11.1%) | |
| 2018 | 4348 | 4203 (10.3%) | 145 (11.5%) | |

Abbreviations: SD, standard deviation; BMI, Body mass index; EQ-5D, EuroQol-5 Dimension

* Chi-Square test or t-test was performed to determine differences between groups with/without stroke. Missing values/nonresponses were excluded from analysis.

* Weighted (%)

stroke without depression group and 2.28 (95% CI 1.91–2.72) in the depression group. Third, the OR for the usual activity domain was 5.09 (95% CI 3.24–8.00) in the stroke with depression group, also higher than 3.12 (95% CI 2.63–3.69) in the stroke without depression group and 2.68 (95%).

**Table 2. Characteristics of stroke population according to presence of depression.**

| Variable | Total | Stroke without depression | Stroke with depression | p-value |
|---|---|---|---|---|
| | n (%) | n (%) | n (%) | |
| **Total** | 1450 | 1321 (91.1%) | 129 (8.9%) | 1450 |
| **Age (mean±SD)** | 66.3±10.2 | 66.4±10.2 | 64.6±10.2 | 0.133 |
| 40–49 | 59 | 52 (6.5%) | 7 (8.5%) | 0.3444 |
| 50–59 | 213 | 195 (20.3%) | 18 (20.2%) | |
| 60–69 | 483 | 429 (29.8%) | 54 (36.2%) | |
| 70–79 | 521 | 477 (31.2%) | 44 (29.5%) | |
| ≥80 | 174 | 168 (12.2%) | 6 (5.6%) | |
| **Gender** | | | | |
| male | 754 | 722 (57.3%) | 32 (24.2%) | <.0001 |
| female | 696 | 599 (42.7%) | 97 (75.8%) | |
| **Region** | | | | |
| Urban area | 1023 | 924 (75.4%) | 99 (83.6%) | 0.0213 |
| Rural area | 427 | 397 (24.6%) | 30 (16.4%) | |
| **Marital status** | | | | |
| Married-cohabit | 1009 | 934 (70.0%) | 75 (59.6%) | 0.0672 |
| Married-no co habit or bereaved or divorced | 14 | 12 (0.8%) | 2 (1.6%) | |
| Unmarried | 426 | 374 (29.2%) | 52 (38.8%) | |
| **Income** | | | | |
| Low | 680 | 615 (44.8%) | 65 (44.2%) | 0.1093 |
| Lower middle | 364 | 330 (24.6%) | 34 (30.0%) | |
| Higher middle | 221 | 200 (16.2%) | 21 (19.8%) | |
| High | 169 | 161 (14.4%) | 8 (6.0%) | |
| **Employment** | | | | |
| Unemployed | 995 | 885 (65.5%) | 110 (86.2%) | <.0001 |
| Employed | 448 | 429 (34.5%) | 19 (13.8%) | |
| **Education Level** | | | | |
| Elementary school or less | 818 | 734 (51.6%) | 84 (59.7%) | 0.3670 |
| Middle school | 245 | 227 (18.5%) | 18 (15.4%) | |
| High school | 266 | 246 (20.2%) | 20 (19.9%) | |
| College or over | 114 | 108 (9.8%) | 6 (5.0%) | |
| **BMI (kg/m$^2$) (mean±SD)** | 24.5± 3.3 | 24.5± 3.2 | 24.9± 3.4 | 0.228 |
| Underweight <18.5 | 36 | 35 (2.3%) | 1 (0.6%) | 0.4675 |
| Normal (18.5–24.9) | 811 | 741 (56.4%) | 70 (56.8%) | |
| Overweight and Obese (≥25.0) | 589 | 531 (41.2%) | 58 (42.6%) | |
| **Alcohol consumption** | | | | |
| Non-drinker | 728 | 649 (46.5%) | 79 (61.0%) | 0.0053 |
| ≤1 drink/mo | 278 | 252 (19.2%) | 26 (22.0%) | |
| 2 drinks/mo to 3 drinks/wk | 311 | 294 (24.7%) | 17 (12.7%) | |
| ≥4 drinks/wk | 121 | 115 (9.6%) | 6 (4.3%) | |
| **Smoking** | | | | |
| Nonsmoker | 721 | 637 (46.2%) | 84 (67.6%) | 0.0003 |
| EX-smoker | 380 | 360 (30.4%) | 20 (16.3%) | |
| Current smoker | 333 | 311 (23.5%) | 22 (16.2%) | |
| **Physical activity (walking)** | | | | |
| No | 377 | 337 (25.1%) | 40 (30.2%) | 0.2993 |
| Yes | 1061 | 972 (74.9%) | 89 (69.8%) | |

*(Continued)*

**Table 2.** (Continued)

| Variable | Total | Stroke without depression | Stroke with depression | p-value |
|---|---|---|---|---|
| | n (%) | n (%) | n (%) | |
| **Days of strength exercise** | | | | |
| No | 1165 | 1056 (78.9%) | 109 (87.3%) | 0.0388 |
| Yes | 276 | 256 (21.1%) | 20 (12.7%) | |
| **Stress level** | | | | |
| Low | 1082 | 1013 (79.0%) | 69 (53.7%) | <.0001 |
| High | 351 | 293 (21.0%) | 58 (46.3%) | |
| **Heath problems (EQ-5D)** | | | | |
| **Mobility** | | | | |
| No | 684 | 636 (51.7%) | 48 (38.9%) | 0.0174 |
| Yes | 766 | 685 (48.3%) | 81 (61.1%) | |
| **Self-care** | | | | |
| No | 1059 | 975 (75.6%) | 84 (63.1%) | 0.0070 |
| Yes | 391 | 346 (24.4%) | 45 (36.9%) | |
| **Usual activity** | | | | |
| No | 842 | 783 (61.5%) | 59 (46.3%) | 0.0031 |
| Yes | 608 | 538 (38.5%) | 70 (53.7%) | |
| **Pain/discomfort** | | | | |
| No | 706 | 663 (51.7%) | 43 (33.6%) | 0.0007 |
| Yes | 744 | 658 (48.3%) | 86 (66.4%) | |
| **Anxiety/depression** | | | | |
| No | 1082 | 1016 (77.9%) | 66 (51.9%) | <.0001 |
| Yes | 368 | 305 (22.1%) | 63 (48.1%) | |
| EQ-5D total | 0.8± 0.2 | 0.8± 0.2 | 0.7± 0.3 | <.0001 |
| **Cycle (year)** | | | | |
| 2008 | 161 | 147 (8.7%) | 14 (7.0%) | 0.6667 |
| 2009 | 146 | 133 (7.4%) | 13 (8.0%) | |
| 2010 | 107 | 99 (6.4%) | 8 (5.1%) | |
| 2011 | 131 | 122 (8.6%) | 9 (7.7%) | |
| 2012 | 93 | 88 (6.3%) | 5 (5.5%) | |
| 2013 | 143 | 133 (11.8%) | 10 (5.7%) | |
| 2014 | 127 | 116 (8.7%) | 11 (7.4%) | |
| 2015 | 129 | 114 (9.8%) | 15 (12.1%) | |
| 2016 | 126 | 115 (10.0%) | 11 (14.4%) | |
| 2017 | 142 | 127 (11.0%) | 15 (11.9%) | |
| 2018 | 145 | 127 (11.2%) | 18 (15.2%) | |

Abbreviations: SD, standard deviation; BMI, Body mass index; EQ-5D, EuroQol-5 Dimension

* Chi-Square test or t-test was performed to determine differences between groups with/without stroke. Missing values/nonresponses were excluded from analysis.

* Weighted (%)

CI 2.34–3.06) in the depression group. Fourth, the OR for the pain/discomfort domain was 3.39 (95% CI 2.15–5.33) in the stroke with depression group, higher than 1.95 in the stroke without depression group and 2.10 in the depression group. Finally, the OR for the anxiety/depression domain was 4.92 (95% CI 3.28–7.39) in the stroke with depression group, higher than 1.83 in the stroke without depression group and 4.56 in the depression group, showing that stroke with depression is more strongly associated with greater anxiety/depression. The

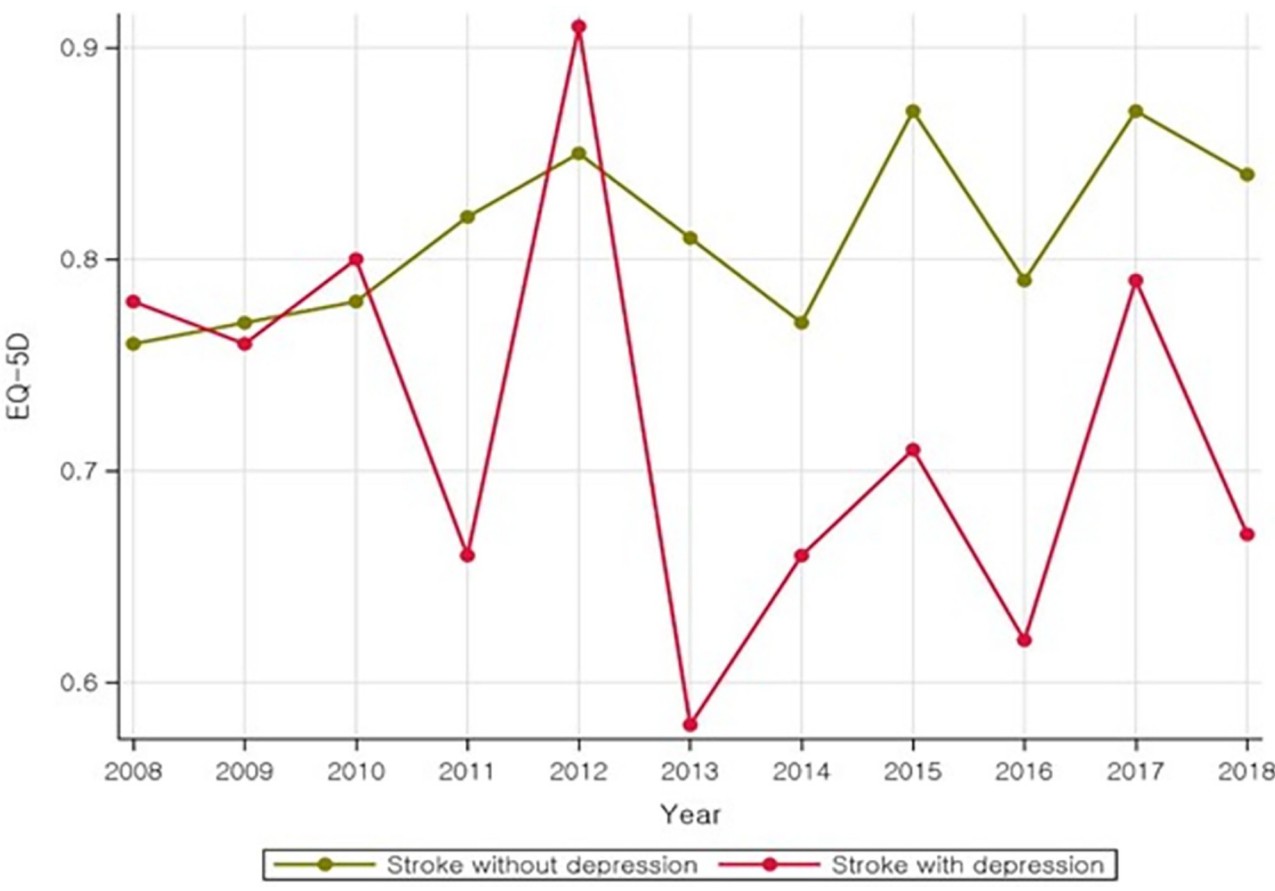

**Fig 2. Prevalence cycle of having total EQ-5D scores in stroke patients with/without depression.**

total EQ-5D score was also lower in the stroke with depression group (-0.16) than in the stroke without depression group (-0.08) and depression group (-0.06), showing that the QoL is the poorest when both diseases are present.

In Model 3, adjusted for demographic, health-related, and disease-related factors, the OR for "having problems in the mobility domain" of the EQ-5D with reference to the no stroke-no depression group was 2.92 (95% CI 1.72–4.96) in the stroke with depression group, higher than 2.31(95% CI 1.95–2.74) in the stroke without depression group and 1.87 (95% CI 1.64–2.13) in the depression group. Likewise, the OR for the self-care domain was 5.12 (95% CI 3.09–8.50) in the stroke with depression group, higher than 3.44 (95% CI 2.84–4.18) in the stroke without depression group and 1.94 (95% CI, 1.62–2.33) in the depression group. Third, the OR for the usual activity domain was 4.27 (95% CI 2.67–6.85) in the stroke with depression group, also higher than 3.10 (95% CI 2.61–3.69) in the stroke without depression group and 2.30 (95% CI 2.00–2.63) in the depression group. Fourth, the OR for the pain/discomfort domain was 2.82 (95% CI 1.75–4.56) in the stroke with depression group, higher than 1.87 in the stroke without depression group and 1.84 in the depression group. Finally, the OR for the anxiety/depression domain was 4.11 in the stroke with depression group, showing a stronger association when both diseases are present. The total EQ-5D score was -0.15 in the stroke with depression group compared to the no stroke-no depression group, while the total EQ-5D scores in the stroke without depression group and depression group were -0.07 and -0.05, respectively, showing that patients with stroke and depression have the poorest QoL (Table 3).

**Table 3. The relationship between quality of life according to stroke and/or depression symptoms using multiple logistic regression.**

| | No Stroke No Depression | Stroke No Depression | | No Stroke Depression | | Stroke Depression | | p for trend | |
|---|---|---|---|---|---|---|---|---|---|
| | | OR(95% CI)[a] | P value | OR(95% CI)[a] | P value | OR(95% CI)[a] | P value | OR(95% CI)[a] | P value |
| **Model 1** | | | | | | | | | |
| Mobility | 1 | 4.88(4.26, 5.59) | <.0001 | 2.65(2.38, 2.95) | <.0001 | 8.22(5.43, 12.4) | <.0001 | 1.85(1.76, 1.94) | <.0001 |
| Self-care | 1 | 7.47(6.38, 8.75) | <.0001 | 3.03(2.58, 3.58) | <.0001 | 13.5(8.93, 20.4) | <.0001 | 2.08(1.96, 2.22) | <.0001 |
| Usual activity | 1 | 6.18(5.37, 7.10) | <.0001 | 3.49(3.09, 3.93) | <.0001 | 11.5(7.68, 17.2) | <.0001 | 2.12(2.01, 2.23) | <.0001 |
| Pain/discomfort | 1 | 2.91(2.53, 3.33) | <.0001 | 2.75(2.47, 3.07) | <.0001 | 6.16(4.01, 9.46) | <.0001 | 1.77(1.68, 1.86) | <.0001 |
| Anxiety/depression | 1 | 2.57(2.18, 3.03) | <.0001 | 5.94(5.31, 6.65) | <.0001 | 8.39(5.68, 12.4) | <.0001 | 2.40(2.28, 2.53) | <.0001 |
| EQ-5D total [a] | 0 | -0.12 | <.0001 | -0.09 | <.0001 | -0.23 | <.0001 | -0.06 | <.0001 |
| Model 2 | | | | | | | | | |
| Mobility | 1 | 2.42(2.06, 2.86) | <.0001 | 2.06(1.81, 2.35) | <.0001 | 3.60(2.22, 5.85) | <.0001 | 1.53(1.44, 1.62) | <.0001 |
| Self-care | 1 | 3.53(2.93, 4.26) | <.0001 | 2.28(1.91, 2.72) | <.0001 | 6.00(3.75, 9.61) | <.0001 | 1.72(1.60, 1.84) | <.0001 |
| Usual activity | 1 | 3.12(2.63, 3.69) | <.0001 | 2.68(2.34, 3.06) | <.0001 | 5.09(3.24, 8.00) | <.0001 | 1.76(1.66, 1.87) | <.0001 |
| Pain/discomfort | 1 | 1.95(1.68, 2.27) | <.0001 | 2.10(1.88, 2.35) | <.0001 | 3.39(2.15, 5.33) | <.0001 | 1.50(1.42, 1.58) | <.0001 |
| Anxiety/depression | 1 | 1.83(1.53, 2.19) | <.0001 | 4.56(4.05, 5.13) | <.0001 | 4.92(3.28, 7.39) | <.0001 | 2.06(1.96, 2.18) | <.0001 |
| EQ-5D total [a] | 1 | -0.08 | <.0001 | -0.06 | <.0001 | -0.16 | <.0001 | -0.04 | <.0001 |
| Model 3 | | | | | | | | | |
| Mobility | 1 | 2.31(1.95, 2.74) | <.0001 | 1.87(1.64, 2.13) | <.0001 | 2.92(1.72, 4.96) | <.0001 | 1.45(1.37, 1.54) | <.0001 |
| Self-care | 1 | 3.44(2.84, 4.18) | <.0001 | 1.94(1.62, 2.33) | <.0001 | 5.12(3.09, 8.50) | <.0001 | 1.61(1.49, 1.73) | <.0001 |
| Usual activity | 1 | 3.10(2.61, 3.69) | <.0001 | 2.30(2.00, 2.63) | <.0001 | 4.27(2.67, 6.85) | <.0001 | 1.65(1.55, 1.75) | <.0001 |
| Pain/discomfort | 1 | 1.87(1.60, 2.19) | <.0001 | 1.84(1.65, 2.07) | <.0001 | 2.82(1.75, 4.56) | <.0001 | 1.41(1.33, 1.48) | <.0001 |
| Anxiety/depression | 1 | 1.91(1.58, 2.31) | <.0001 | 3.67(3.22, 4.18) | <.0001 | 4.11(2.56, 6.60) | <.0001 | 1.88(1.77, 2.00) | <.0001 |
| EQ-5D total [a] | 0 | -0.07 | <.0001 | -0.05 | <.0001 | -0.15 | <.0001 | -0.03 | <.0001 |

Abbreviations: OR, odds ratio; CI, confidence interval; EQ-5D, EuroQol-5 Dimension

[a] β coefficient in linear regression analysis.

Model 1 was unadjusted

Model 2 was adjusted for age, gender, region, marital status, income, employment, education

Model 3 was adjusted for age, gender, region, marital status, income, employment, education, BMI, alcohol consumption, smoking, physical activity, days of strength exercise, stress level

## Logistic regression for the association of stroke and depression treatment with QoL

Table 4 shows the relationships of no stroke or depression treatment group (control group), stroke treatment group, depression treatment group, and concurrent stroke and depression treatment group with each of the domains of QoL analyzed using logistic regression.

In model 2 that was adjusted for demographic factors, with reference to the control group, the OR for the mobility domain of the EQ-5D was 4.31 (95% CI 2.13–8.74) in the concurrent stroke and depression treatment group, higher than 2.60 (95% CI 2.14–3.15) in the stroke treatment group and 2.38 (95% CI 1.96–2.90) in the depression treatment group. The OR for the self-care domain was 4.19 (95% CI 3.39–5.18) in the stroke treatment group and 2.28 (95% CI 1.77–2.93) in the depression treatment group with reference to the concurrent stroke and depression treatment group, showing strong associations. Third, the OR for "having problems in the usual activity domain" was 5.64 (95% CI 2.94–10.8) in the concurrent stroke and depression treatment group with reference to the control group, which was higher than 3.54 (95% CI 2.91–4.31) in the stroke treatment group and 3.41 (95% CI 2.78–4.19) in the depression treatment group. Fourth, the OR for "having problems in the pain/discomfort domain" was also

**Table 4. The relationship between quality of life according to treatment of stroke and/or depression symptoms using multiple logistic regression.**

| | No Stroke treatment No Depression treatment | Stroke treatment No Depression treatment | | No Stroke treatment Depression treatment | | Stroke treatment Depression treatment | | p for trend | |
|---|---|---|---|---|---|---|---|---|---|
| | | OR(95% CI)[a] | P value | OR(95% CI)[a] | P value | OR(95% CI)[a] | P value | OR(95% CI)[a] | P value |
| **Model 1** | | | | | | | | | |
| Mobility | 1 | 5.11(4.36, 6.00) | <.0001 | 3.42(2.89, 4.04) | <.0001 | 10.8(5.99, 19.4) | <.0001 | 2.24(2.08, 2.41) | <.0001 |
| Self-care | 1 | 8.78(7.37, 10.5) | <.0001 | 3.79(3.00, 4.77) | <.0001 | 19.8(11.2, 34.9) | <.0001 | 2.58(2.38, 2.80) | <.0001 |
| Usual activity | 1 | 6.87(5.83, 8.09) | <.0001 | 5.03(4.24, 5.98) | <.0001 | 13.9(7.91, 24.3) | <.0001 | 2.70(2.51, 2.91) | <.0001 |
| Pain/discomfort | 1 | 2.98(2.53, 3.50) | <.0001 | 3.32(2.80, 3.94) | <.0001 | 7.45(4.21, 13.2) | <.0001 | 2.00(1.86, 2.16) | <.0001 |
| Anxiety/depression | 1 | 2.46(2.03, 2.97) | <.0001 | 9.31(7.86, 11.0) | <.0001 | 13.2(7.28, 23.8) | <.0001 | 2.90(2.69, 3.13) | <.0001 |
| EQ-5D total [a] | 0 | -0.14 | <.0001 | -0.13 | <.0001 | -0.28 | <.0001 | -0.08 | <.0001 |
| **Model 2** | | | | | | | | | |
| Mobility | 1 | 2.60(2.14, 3.15) | <.0001 | 2.38(1.96, 2.90) | <.0001 | 4.31(2.13, 8.74) | <.0001 | 1.71(1.57, 1.85) | <.0001 |
| Self-care | 1 | 4.19(3.39, 5.18) | <.0001 | 2.28(1.77, 2.93) | <.0001 | - | - | 1.92(1.75, 2.11) | <.0001 |
| Usual activity | 1 | 3.54(2.91, 4.31) | <.0001 | 3.41(2.78, 4.19) | <.0001 | 5.64(2.94, 10.8) | <.0001 | 2.07(1.89, 2.26) | <.0001 |
| Pain/discomfort | 1 | 2.01(1.68, 2.40) | <.0001 | 2.39(1.98, 2.87) | <.0001 | 4.21(2.16, 8.21) | <.0001 | 1.62(1.50, 1.76) | <.0001 |
| Anxiety/depression | 1 | 1.76(1.42, 2.17) | <.0001 | 6.68(5.54, 8.05) | <.0001 | 7.34(3.79, 14.2) | <.0001 | 2.39(2.20, 2.60) | <.0001 |
| EQ-5D total [a] | 1 | -0.09 | <.0001 | -0.09 | <.0001 | -0.21 | <.0001 | -0.06 | <.0001 |
| **Model 3** | | | | | | | | | |
| Mobility | 1 | 2.46(2.00, 3.02) | <.0001 | 2.02(1.65, 2.46) | <.0001 | 3.49(1.57, 7.77) | 0.0022 | 1.58(1.45, 1.72) | <.0001 |
| Self-care | 1 | 4.03(3.23, 5.04) | <.0001 | 1.78(1.38, 2.29) | <.0001 | - | - | 1.75(1.59, 1.92) | <.0001 |
| Usual activity | 1 | 3.48(2.83, 4.26) | <.0001 | 2.63(2.14, 3.24) | <.0001 | 5.04(2.44, 10.4) | <.0001 | 1.87(1.71, 2.04) | <.0001 |
| Pain/discomfort | 1 | 1.90(1.58, 2.29) | <.0001 | 1.95(1.61, 2.35) | <.0001 | 3.60(1.73, 7.49) | 0.0006 | 1.48(1.37, 1.61) | <.0001 |
| Anxiety/depression | 1 | 1.81(1.45, 2.26) | <.0001 | 5.06(4.10, 6.25) | <.0001 | 7.28(3.28, 16.2) | <.0001 | 2.16(1.96, 2.36) | <.0001 |
| EQ-5D total [a] | 0 | -0.08 | <.0001 | -0.08 | <.0001 | -0.2 | <.0001 | -0.05 | <.0001 |

Abbreviations: OR, odds ratio; CI, confidence interval; EQ-5D, EuroQol-5 Dimension

[a] β coefficient in linear regression analysis.

Model 1 was unadjusted

Model 2 was adjusted for age, gender, region, marital status, income, employment, education

Model 3 was adjusted for age, gender, region, marital status, income, employment, education, BMI, alcohol consumption, smoking, physical activity, days of strength exercise, stress level

the highest in the concurrent stroke and depression treatment group, at 4.21 (95% CI 2.16–8.21). Finally, the OR for the anxiety/depression domain was the highest in the concurrent stroke and depression treatment group, at 7.34 (95% CI 3.79–14.2). The total EQ-5D score was the lowest in concurrent stroke and depression treatment group (-0.21) compared to the control group.

In model 3 that additionally adjusted for health-related factors, with reference to the control group, the OR for the mobility domain of the EQ-5D was 3.49 (95% CI 1.57–7.77) in the concurrent stroke and depression treatment group, higher than 2.46 (95% CI 2.00–3.02) in the stroke treatment group and 2.02 (95% CI 1.65–2.46) in the depression treatment group. The OR for the self-care domain was 4.03 (95% CI 3.23–5.04) in the stroke treatment group with reference to the control group, which was higher than 1.78 (95% CI 1.38–2.29) in the depression treatment group. Third, the OR for "having problems in the usual activity domain" was 5.04 (95% CI 2.44–10.4) in the concurrent stroke and depression treatment group with reference to the control group, which was higher than 3.48 (95% CI 2.83–4.26) in the stroke treatment group and 2.63 (95% CI 2.14–3.24) in the depression treatment group. Fourth, the OR for "having problems in the pain/discomfort domain" was also the highest in the concurrent

stroke and depression treatment group, at 3.60 (95% CI 1.73–7.49). Finally, the OR for the anxiety/depression domain was the highest in the concurrent stroke and depression treatment group, at 7.28 (95% CI 3.28–16.2). The total EQ-5D score was also the lowest in concurrent stroke and depression treatment group (-0.20), compared to the control group (Table 4).

## Discussion

This study analyzed the associations between stroke, depression, and HRQoL using data from 45,741 adults aged 40 years and over in the KNHANES IV–VII. The results of this study regarding the associations between stroke, depression, and QoL are in line with previous findings [18–21]. Previously, one study examined the enhancement of the QoL of stroke patients through treatment and rehabilitation strategies [22], and a systematic review revealed that age, sex, fatigue, depression, self-efficacy, and QoL predict physical activity limitations in stroke patients [23]. Several studies have investigated the association between stroke and QoL. In 2015, Ran et al. reported that stroke is directly associated with poor QoL [20]. Our study also confirms that stroke patients have a lower EQ-5D score, indicating a poorer QoL, compared to individuals without stroke, and that stroke strongly predicts problems in the five domains of QoL: mobility, self-care, usual activity, pain/discomfort, anxiety/depression.

Depression has also been extensively studied in the literature. It has been reported that depression in older adults, unemployed, overweight and obese, and mentally ill individuals is closely associated with a low QoL. Depression often develops secondarily to various diseases, and many studies have shown that these individuals have a poorer QoL [18, 19, 24–27]. In this study, we confirmed that patients suffering from concurrent diagnosis of both stroke and depression have a poorer QoL than stroke patients without depression or patients not diagnosed with stroke but with depression.

From 2008–2018, the percentage of stroke patients with depression was on a steady rise. According to Son in 2015, the prevalence of depression in community-dwelling stroke patients (excluding acute-stage patients and patients hospitalized in rehabilitation hospitals) was 14% [28], and Ayerbe et al. reported an incidence of post-stroke depression over 15 years to be 7–21% [29]. These rates are similar to the prevalence of 8.7% in 2008 and 15.2% in 2018 observed in this study, of which the results can be interpreted on a larger population of the national level. Furthermore, this study expanded the understandings on the HRQoL of stroke patients by showing that patients undergoing treatments for depression may still require further attention in their HRQoL in addition to symptom management. While plenty of literature stated the association between stroke, depression, and HRQoL [21, 30–34], there has yet been any study assessing the impairment of HRQoL of stroke patients undergoing treatment for depression. Further studies are necessary to elucidate the specific factors causing this impairment of HRQoL of the stroke patients suffering from depression, which may be psychological, social, or due to biological responses from medications.

One strength of this study is that we divided the participants into four groups according to disease prevalence of stroke and depression and indirectly determined the severity of the conditions based on their treatment status. Through our analyses, we were able to show that stroke patients who are being treated with depression have a poorer QoL. This suggests that, compared to patients who have only been diagnosed and are not being treated, patients who are being treated have more severe impairment in QoL. Moreover, we compared the relationships of each of the domains of EQ-5D with the stroke treatment group, depression treatment group, and concurrent stroke and depression treatment group, and the concurrent stroke and depression treatment group had the most substantial impairment of all of the domains related to HRQoL.

This study has a few limitations. First, although we confirmed that stroke and depression are associated with impaired HRQoL, we could not establish a causality between them because we used cross-sectional data from the KNHANES. In other words, we could not establish whether stroke is an independent causative factor of depression or an epiphenomenon factor of depression. Second, the prevalence of two diseases was surveyed using a self-report questionnaire, and no questions about the severity of the symptoms were used; hence, we could not objectively determine the severity of stroke and depression in this study. Third, the study data may be biased because a self-report questionnaire was used to collect information about the participants' demographics, health-related lifestyle, and mental health. Fourth, the KNHANES is a study conducted in Korea, so researchers should take caution when generalizing or applying the findings of this study to other countries. Finally, stroke and depression were diagnosed based on certain diagnostic criteria, and the diagnostic criteria may show the level of progression of the disease depending on the features of the disease. However, the severity of disease based on such diagnostic criteria was not considered in this study, and we only analyzed data about whether a diagnosis has been made.

Additional studies are required to investigate the causality relationship between stroke, depression, and QoL as well as confounders, and there are a few suggestions for subsequent studies in order to address the limitations of this study. First, studies should classify the severity of symptoms into several levels or examine the effects of varying severity of symptoms, as determined through expert assessments, on QoL. In addition, various other study designs, such as case control studies and cohort studies, should be utilized to substantiate causality. Studies should also employ the national health insurance data and other official data in addition to self-report data to ensure objectivity of data. Moreover, we recommend using scales that objectively assess the causes of stroke and sequelae, such as paralysis, and also considering weekly prevalence rates and confounders.

Further findings on the causation or confounders in the relationship among stroke, depression, and QoL would serve as clinical evidence for stroke and depression.

## Conclusion

This study confirmed that having both stroke and depression is associated with a significant impairment of HRQoL, compared with having neither or having only stroke or depression in Korean adults aged 40 years or older using nationally representative data. Further, we also observed that patients receiving treatment for both stroke and depression had poorer QoL than individuals not being treated for any of the conditions or individuals being treated for only one of the conditions. The results of this study highlight the need for multilateral and comprehensive approaches to stroke and depression and serve as foundational data for developing such approaches.

## Supporting information

**S1 Checklist. STROBE 2007 (v4) statement—Checklist of items that should be included in reports of *cross-sectional studies*.**
(DOCX)

## Author Contributions

**Conceptualization:** Sook-Hyun Lee.

**Data curation:** Sun Woo Kang, Ye-Seul Lee.

**Formal analysis:** Ye-Seul Lee.

**Methodology:** Sook-Hyun Lee.

**Project administration:** In-Hyuk Ha.

**Supervision:** Yoon Jae Lee, In-Hyuk Ha.

**Validation:** Seungwon Kwon, Peggy Bosch, Yoon Jae Lee.

**Writing – original draft:** Sun Woo Kang, Sook-Hyun Lee.

**Writing – review & editing:** Sun Woo Kang, Ye-Seul Lee, Seungwon Kwon, Peggy Bosch, Yoon Jae Lee, In-Hyuk Ha.

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
