## [Decision Letter · Decision Letter 0]

21 Feb 2022

PONE-D-21-40161Association between depression and quality of life in stroke patients: The Korea National Health and Nutrition Examination Survey (KNHANES) IV–VII (2008–2018)PLOS ONE

Dear Dr. Ha,

Thank you for submitting your manuscript to PLOS ONE. After careful consideration, we feel that it has merit but does not fully meet PLOS ONE’s publication criteria as it currently stands. Therefore, we invite you to submit a revised version of the manuscript that addresses the points raised during the review process. The revised version should address all comments. You may also note the indirect costs of mental health in terms of lost earnings: https://doi.org/10.1111/acps.13067

We look forward to receiving your revised manuscript.

Kind regards,

Petri Böckerman

Academic Editor

PLOS ONE

Journal Requirements:

“This research was funded by the Jaseng Medical Foundation, Republic of Korea. (JS-RP-2021-24)”

“This research was funded by the Jaseng Medical Foundation, Republic of Korea. (JS-RP-2021-24)”

We note that you have provided funding information within the Funding Section. Please note that funding information should not appear in the Acknowledgments section or other areas of your manuscript. We will only publish funding information present in the Funding Statement section of the online submission form.

“This research was funded by the Jaseng Medical Foundation, Republic of Korea. (JS-RP-2021-24)”

Reviewers' comments:

Reviewer's Responses to Questions

**Comments to the Author**

1. Is the manuscript technically sound, and do the data support the conclusions?

Reviewer #1: Yes

Reviewer #2: Yes

Reviewer #3: Yes

2. Has the statistical analysis been performed appropriately and rigorously? 

Reviewer #1: Yes

Reviewer #2: Yes

Reviewer #3: Yes

3. Have the authors made all data underlying the findings in their manuscript fully available?

Reviewer #1: Yes

Reviewer #2: Yes

Reviewer #3: Yes

4. Is the manuscript presented in an intelligible fashion and written in standard English?

Reviewer #1: Yes

Reviewer #2: Yes

Reviewer #3: Yes

5. Review Comments to the Author

Reviewer #1: Thank you for inviting me to review this well written paper on the effects of depression and stroke on QoL. There are however some concerns that the author's must address, here are my comments for the author's consideration,

Introduction

Line 76- What psychological responses are triggered? This paragraph is unclear

Methods

Confirm that data on all covariates was extracted through KHANES survey data or was the data collected from patients directly?

Confirm that EQ-5D data was also extracted from the KHANES survey, was the EQ-5D instrument applied to the study population of KHANES survey?

Results

Lines 216-301 The results in the table and text are repetitive, please present only the important findings in text, all the adjusted models need not be explained in text, authors could restrict themselves to presenting the final adjusted model's important results in the text.

Discussion

Line 321-322 - What previous studies? This sentence needs citations

Lines 333-335 - This association has been reported in previously published papers before, for eg. https://pubmed.ncbi.nlm.nih.gov/11150931/. I am unsure as to whether this article bridges a critical gap in the existing literature. The authors could substantiate how the study addresses an existing gap in the literature more in the discussion.

Lines 342-344 - Again many studies have been published on stroke, depression and QoL https://www.ncbi.nlm.nih.gov/pmc/articles/PMC6797138/

https://www.ncbi.nlm.nih.gov/pmc/articles/PMC5900407/

Author's could discuss the study's findings in the context of the existing literature on the topic.

Lines 349-352 - Should be in the methods section of the paper, not the discussion

Lines 378- 379 - Cohort studies have already been done on this area, as stated previously please discuss this articles findings in context of other articles that have been published already.

Reviewer #2: Depression was strongly associated with QoL in stroke patients. This study showed the clinical approaches that take QoL into consideration are needed for stroke patients with depression. Please correct below.

Line 37, explain ‘EQ-5D score’, like ‘Health related quality of life’

Line 85, please change ‘examined this condition.[10-12]’ -> ‘examined this condition [10-12].’

Reviewer #3: Firstly I would like to appreciate the authors for their effort and interesting study. The big strength being the national data in Korea, the findings are great.

1. This study requires to describe more detail in Materials and Methods Section (Line 100). Please describe the summary of the KNHANES IV-VII in order to share knowledge to readers, and sampling procedure will be more clear. Sampling procedure should be described step by step.

2. In Line 101, 93,028 participants were adults or either needed to describe.

3. Line 101, study participants were described as 40 years or older, but in Abstract, <40 years, it may be typing error, but really important point that should be careful.

4. Line 31-32, "The participants were divided into stroke and non-stroke group" may make the reader confuse. This study included the stroke patients with or without depression, right? Again sampling should be described step by step.

5. Line 34, incidence or prevalence?

6. Line 41, "more severe condition" makes confusion - the study didn't assess the severity of either depression or stroke. This fact is also discussed in limitation.

7. In Conclusion (Line 387-388), "having both stroke and depression has a greater impact on HRQoL", could you please describe the impact as positive or negative or something clearer words? Again, "neither innLine 388" may confuse non-stroke patients were included in this study.

8. Line 390, those received treatment for both stroke and depression had poorer QoL. It may be due to they had severe disease for stroke or depression. The authors may need to elaborate in discussion section together with other relevant previous studies.

9. Line 73-83, not relevant to explain in this study, I think. It would be better to elaborate more on public health problems related to depression in stroke patients.

10. Line 123, "obesity 25 and above"- please revise according to WHO that obese defines for BMI 30.0 and above.

11.Line 127-131, definitions of stroke and depression are not clear enough. Just defined when a diagnosis by physician? How physician defined the diagnosis? KNHANES may include those definitions that the physician define by 1,2,3,.... Could you please improve the definitions?

12. Line 133, EQ-5D tools was used. Why the authors chose this tool. Why not WHO QoL tools? The authors may give detail explanation or justification why EQ-5D is used or EQ-5D is the most appropriate tool for this study. please elaborate or summarize the tool, what is EQ-5D and components, how it measures, and what are the advantages to use this EQ-5D, and so on.

13. Line 30, 45,741 or Line 320, 44,291. Final participants in this study, which one is right?

14. Line 335, non-stroke patients? Are they included?

15. Line 345-348, I think these are not strengths.

16. 349-352, should describe in Materials and Methods section.

17. Line 358, "strongest association", positive or negative or something should be described here.

6. PLOS authors have the option to publish the peer review history of their article (what does this mean?). If published, this will include your full peer review and any attached files.

Reviewer #1: **Yes: **Nesa Aurlene

Reviewer #2: No

Reviewer #3: No

---

## [Author Response · Author response to Decision Letter 0]

14 Apr 2022

Reviewer #1: Thank you for inviting me to review this well written paper on the effects of depression and stroke on QoL. There are however some concerns that the author's must address, here are my comments for the author's consideration,

Introduction

Line 76- What psychological responses are triggered? This paragraph is unclear.

- We appreciate the reviewer’s comment. Based on the reviewer’s comment, we revised the manuscript and clarified the paragraph as follows:

One of the consequences of stroke such as the newly acquired physical disabilities and subsequent social isolation caused by stroke, trigger psychological responses such as anger and despair; in addition, physical inactivity and loss of physical sensations due to physical disabilities also cause depression. These findings imply the substantial impairment of QoL by post-stroke depression. Moreover, post-stroke depression aggravates the burden on caregivers [8]. 

Methods

Confirm that data on all covariates was extracted through KHANES survey data or was the data collected from patients directly?

Confirm that EQ-5D data was also extracted from the KHANES survey, was the EQ-5D instrument applied to the study population of KHANES survey?

- We appreciate the reviewer’s comment. EQ-5D was extracted from the KHANES survey data. Based on the reviewer’s comment, we revised the manuscript as follows:

Health-related QoL (HRQoL) was assessed using the EQ-5D, which was included as a part of the KNHANES survey.

Results

Lines 216-301 The results in the table and text are repetitive, please present only the important findings in text, all the adjusted models need not be explained in text, authors could restrict themselves to presenting the final adjusted model's important results in the text.

- We appreciate the reviewer’s comment. Based on the reviewer’s comment, we revised the manuscript and revised the paragraph as follows:

Table 3 shows the association of each of the five domains of QoL with four groups of Korean adults aged 40 years and over: no stroke-no depression group, stroke without depression group, depression group, and stroke with depression group. In Model 2 adjusted for demographic factors, the OR for the mobility domain of the EQ-5D with reference to the no stroke-no depression group was 3.60 (95% CI 2.22, 5.85) in the stroke with depression group, higher than 2.42 (95% CI 2.06, 2.86) in the stroke without depression group and 2.06 (95% CI 1.81, 2.35) in the depression group.

In Model 3, adjusted for demographic, health-related, and disease-related factors, the OR for “having problems in the mobility domain” of the EQ-5D with reference to the no stroke-no depression group was 2.92

Discussion

Line 321-322 - What previous studies? This sentence needs citations

- We appreciate the reviewer’s comment. Based on the reviewer’s comment, we revised the manuscript and revised the sentence as follows:

The results of this study regarding the associations between stroke, depression, and QoL are in line with previous findings[19-22].

Lines 333-335 - This association has been reported in previously published papers before, for eg. https://pubmed.ncbi.nlm.nih.gov/11150931/. I am unsure as to whether this article bridges a critical gap in the existing literature. The authors could substantiate how the study addresses an existing gap in the literature more in the discussion.

- We appreciate the reviewer’s comment. As the reviewer suggested, previous literature supports the current findings on the association between stroke, depression and QoL. Furthermore, this study expanded this finding by comparing this with not only stroke patients without depression, but also patients without stroke but with depression, using a nationally representative database. Moreover, this study identified patients undergoing treatment for both stroke and depression and showed that those with concurrent treatments for both diseases had a more severe impairment in HRQoL. Therefore, we concluded from this study that first, this finding is generalizable to the national population; and second, this study shows that patients under treatment require further monitoring on their HRQoL.

Based on the reviewer’s comment, we revised the manuscript as the following:

Discussion

These rates are similar to the prevalence of 8.7% in 2008 and 15.2% in 2018 observed in this study, of which the results can be interpreted on a larger population of the national level. Furthermore, this study expanded the understandings on the HRQoL of stroke patients by showing that patients undergoing treatments for depression may still require further attention in their HRQoL in addition to symptom management. While plenty of literature stated the association between stroke, depression, and HRQoL [21, 30-34], there has yet been any study assessing the impairment of HRQoL of stroke patients undergoing treatment for depression. Further studies are necessary to elucidate the specific factors causing this impairment of HRQoL of the stroke patients suffering from depression, which may be psychological, social, or due to biological responses from medications.

Lines 342-344 - Again many studies have been published on stroke, depression and QoL https://www.ncbi.nlm.nih.gov/pmc/articles/PMC6797138/

https://www.ncbi.nlm.nih.gov/pmc/articles/PMC5900407/

Author's could discuss the study's findings in the context of the existing literature on the topic.

- We appreciate the reviewer’s comment. Based on the reviewer’s comment, we revised the manuscript as the following:

Discussion

These rates are similar to the prevalence of 8.7% in 2008 and 15.2% in 2018 observed in this study, of which the results can be interpreted on a larger population of the national level. Furthermore, this study expanded the understandings on the HRQoL of stroke patients by showing that patients undergoing treatments for depression may still require further attention in their HRQoL in addition to symptom management. While plenty of literature stated the association between stroke, depression, and HRQoL [21, 30-34], there has yet been any study assessing the impairment of HRQoL of stroke patients undergoing treatment for depression. Further studies are necessary to elucidate the specific factors causing this impairment of HRQoL of the stroke patients suffering from depression, which may be psychological, social, or due to biological responses from medications.

Lines 378- 379 - Cohort studies have already been done on this area, as stated previously please discuss this articles findings in context of other articles that have been published already.

- We appreciate the reviewer’s comment. Based on the reviewer’s comment, we revised the manuscript as the following:

Discussion

These rates are similar to the prevalence of 8.7% in 2008 and 15.2% in 2018 observed in this study, of which the results can be interpreted on a larger population of the national level. Furthermore, this study expanded the understandings on the HRQoL of stroke patients by showing that patients undergoing treatments for depression may still require further attention in their HRQoL in addition to symptom management. While plenty of literature stated the association between stroke, depression, and HRQoL [21, 30-34], there has yet been any study assessing the impairment of HRQoL of stroke patients undergoing treatment for depression. Further studies are necessary to elucidate the specific factors causing this impairment of HRQoL of the stroke patients suffering from depression, which may be psychological, social, or due to biological responses from medications.

 

Reviewer #2: Depression was strongly associated with QoL in stroke patients. This study showed the clinical approaches that take QoL into consideration are needed for stroke patients with depression. Please correct below.

Line 37, explain ‘EQ-5D score’, like ‘Health related quality of life’

- We appreciate the reviewer’s comment. Based on the reviewer’s comment, we revised the manuscript and revised the abstract, line 37 as follows:

Results The overall incidence of stroke was 3.2%, and the incidence was 9% higher in men than in women. Multiple logistic regression was performed after adjusting for demographic factors, health-related factors, and disease-related factors. The results confirmed that the stroke group with depression had a lower overall health-related quality of life, measured using EQ-5D, score compared to the stroke group without depression (-0.15).

Line 85, please change ‘examined this condition.[10-12]’ -> ‘examined this condition [10-12].’

- We appreciate the reviewer’s comment. Based on the reviewer’s comment, we revised the manuscript as below:

As a result of the rising prevalence of stroke worldwide, the prevalence of post-stroke depression is also increasing, and many studies have examined this condition[8-10].  

Reviewer #3: Firstly I would like to appreciate the authors for their effort and interesting study. The big strength being the national data in Korea, the findings are great.

1. This study requires to describe more detail in Materials and Methods Section (Line 100). Please describe the summary of the KNHANES IV-VII in order to share knowledge to readers, and sampling procedure will be more clear. Sampling procedure should be described step by step.

- We appreciate the reviewer’s comment. Based on the reviewer’s comment, we revised the manuscript and added the following paragraph:

Data from the KNHANES IV–VII, conducted from January 2008 to December 2018, was used in this study. KNHANES database is built based on a complex sample survey for which the national population is sampled using three-stage cluster stratification method. Stratification variables for building this database included principal administrative regions and type of residence in the first strata; type of household and household member characteristics in the second strata; and subdivisions in the administrative regions in the third strata[15]. For the analysis in this study, sample weights, variance strata, and stratification variables were applied to obtain representativeness on the national population.

2. In Line 101, 93,028 participants were adults or either needed to describe.

- We really appreciate the reviewer’s comment. The initial sample of 93,028 participants in KNHANES survey included participants across all ages. We revised the manuscript as below:

In KNHANES IV-VII, in which 93,028 participants across all ages were surveyed, we extracted the data of 45,741 adults aged 40 years or older who participated in the health examination.

3. Line 101, study participants were described as 40 years or older, but in Abstract, <40 years, it may be typing error, but really important point that should be careful.

- We really appreciate the reviewer’s comment. As the reviewer pointed out, the Abstract need to be changed. We revised the manuscript as below:

adults who were aged >40 years and had no missing data for stroke and depression were included in the analysis.

4. Line 31-32, "The participants were divided into stroke and non-stroke group" may make the reader confuse. This study included the stroke patients with or without depression, right? Again sampling should be described step by step.

- We appreciate the reviewer’s comment and agree that current wording was somewhat confusing. This study divided the participants into four groups, by the prevalence of both stroke and depression. We revised the manuscript as below:

Abstract

The participants were first grouped by prevalence of stroke, and further divided by prevalence of depression.

Methods

From 45,741 adults included in this study, the participants were first grouped by prevalence of stroke, and further subdivided by prevalence of depression.

5. Line 34, incidence or prevalence?

- We appreciate the reviewer’s comment. We revised the manuscript as below:

Results The overall prevalence of stroke was 3.2%,

6. Line 41, "more severe condition" makes confusion - the study didn't assess the severity of either depression or stroke. This fact is also discussed in limitation.

- We appreciate the reviewer’s comment. We revised the manuscript as below:

Conclusion Depression was strongly associated with QoL in stroke patients. This association was more evident in stroke patients undergoing treatment for depression.

Discussion

This suggests that, compared to patients who have only been diagnosed and are not being treated, patients who are being treated have more severe impairment in QoL.

7. In Conclusion (Line 387-388), "having both stroke and depression has a greater impact on HRQoL", could you please describe the impact as positive or negative or something clearer words? Again, "neither innLine 388" may confuse non-stroke patients were included in this study.

- We appreciate the reviewer’s comment. As we have explained in the previous comment, this study included four groups which were divided by prevalence of stroke and depression. Therefore, we revised the manuscript as below to convey the negative impact on HRQoL:

Conclusion

This study confirmed that having both stroke and depression is associated with a significant impairment of HRQoL,

8. Line 390, those received treatment for both stroke and depression had poorer QoL. It may be due to they had severe disease for stroke or depression. The authors may need to elaborate in discussion section together with other relevant previous studies.

- We appreciate the reviewer’s comment. Based on the reviewer’s comment, we revised the manuscript as below:

Discussion

These rates are similar to the prevalence of 8.7% in 2008 and 15.2% in 2018 observed in this study, of which the results can be interpreted on a larger population of the national level. Furthermore, this study expanded the understandings on the HRQoL of stroke patients by showing that patients undergoing treatments for depression may still require further attention in their HRQoL in addition to symptom management. While plenty of literature stated the association between stroke, depression, and HRQoL [21, 30-34], there has yet been any study assessing the impairment of HRQoL of stroke patients undergoing treatment for depression. Further studies are necessary to elucidate the specific factors causing this impairment of HRQoL of the stroke patients suffering from depression, which may be psychological, social, or due to biological responses from medications.

9. Line 73-83, not relevant to explain in this study, I think. It would be better to elaborate more on public health problems related to depression in stroke patients.

- We appreciate the reviewer’s comment. Based on the reviewer’s comment, we revised the paragraph as below:

One of the consequences of stroke such as the newly acquired physical disabilities and subsequent social isolation caused by stroke, trigger psychological responses such as anger and despair; in addition, physical inactivity and loss of physical sensations due to physical disabilities also cause depression. These findings imply the substantial impairment of QoL by post-stroke depression. Moreover, post-stroke depression aggravates the burden on caregivers [8]. 

10. Line 123, "obesity 25 and above"- please revise according to WHO that obese defines for BMI 30.0 and above.

- We appreciate the reviewer’s comment. Based on the reviewer’s comment, we revised the sentence as below:

based on the WHO criteria, it was classified into underweight (<18.5 kg/m2), normal (18.5 to <25 kg/m2), and overweight and obese (≥25 kg/m2) [31].

11. Line 127-131, definitions of stroke and depression are not clear enough. Just defined when a diagnosis by physician? How physician defined the diagnosis? KNHANES may include those definitions that the physician define by 1,2,3,…. Could you please improve the definitions?

- We appreciate the reviewer’s comment. As the reviewer pointed out, the definition of stroke and depression may seem vague at the current state. 

The KNHANES survey items regarding the diagnosis of stroke and depression consisted of four questions, respectively, in a similar format: “A. had the participant ever been diagnosed with the disease (stroke or depression)?” “B. if so, when was this diagnosis made?” “C. is the participant diagnosed with stroke/depression within the two-weeks period before the answering the survey?” And “D. if the participant answered that he/she has history of the disease (stroke or depression), had he/she received any kind of medical treatment?” 

Since the survey was conducted to each participant from the patient’s point of view, the participant is asked to provide a proof for the medical diagnosis that he/she stated, for example with the receipt of their most recent hospital visits or prescriptions. However, KNHANES survey does not include the diagnosis guidelines for each medical decisions, e.g. MRIs, blood tests, or DSM-IV criteria. 

Based on the reviewer’s comment, we revised the manuscript as below:

KNHANES survey items regarding the patients’ diagnosis history of stroke and depression were adopted to define the diseases in this study. Since this survey was conducted on each participant who were answering from the patient’s point of view, each item was answered within the range of yes or no, and no specific details were included for each diagnosis. The survey item on the medical diagnosis of the disease within the two-weeks period before answering the survey was used to define diagnosis of stroke and/or depression of the patient. The survey item on the history of medical treatments for stroke and/or depression was used to define treatment history of stroke and/or depression.

12. Line 133, EQ-5D tools was used. Why the authors chose this tool. Why not WHO QoL tools? The authors may give detail explanation or justification why EQ-5D is used or EQ-5D is the most appropriate tool for this study. please elaborate or summarize the tool, what is EQ-5D and components, how it measures, and what are the advantages to use this EQ-5D, and so on.

- We appreciate the reviewer’s comment. As the reviewer pointed out, WHO-QoL tools are also available to measure HRQoL, and in some circumstances it would be more efficient to use WHO-QoL instead of EQ-5D tools. This study employed KNHANES survey database, which has already adopted EQ-5D as part of their survey throughout all period of the survey and across all samples. Therefore, this study had only one choice in measuring the HRQoL of participants.

Based on the reviewer’s comment, we revised manuscript as the following:

Health-related QoL (HRQoL) was assessed using the EQ-5D, which was included as part of the KNHANES survey. Since this tool was officially included in KNHANES survey, it was possible to compare HRQoL across different patient groups within the participants. The EQ-5D evaluates HRQoL based on mobility, self-care, usual activities, pain/discomfort, and anxiety/depression, and it is a widely employed tool for assessing HRQoL across different disease states, allowing for comparisons across patient groups and diseases.

13. Line 30, 45,741 or Line 320, 44,291. Final participants in this study, which one is right?

- Thank you for the reviewer’s comment. The number 45,741 is correct and we revised the manuscript as below:

Discussion

This study analyzed the associations between stroke, depression, and HRQoL using data from 45,741 adults aged 40 years and over in the KNHANES IV–VII.

14. Line 335, non-stroke patients? Are they included?

- We appreciate the reviewer’s comment. As we have elaborated earlier, this study analyzes the four groups which are divided by the diagnosis of stroke and depression. The results showed that patients with both stroke and depression have a lower HRQoL than those with only stroke or those with only depression. Based on the reviewer’s comment, we revised the manuscript as below:

In this study, we confirmed that patients suffering from concurrent diagnosis of both stroke and depression have a poorer QoL than stroke patients without depression or patients not diagnosed with stroke but with depression. 

15. Line 345-348, I think these are not strengths.

- We humbly appreciate the reviewer’s comment and revised the manuscript as below:

One strength of this study is that we divided the participants into four groups according to disease prevalence of stroke and depression and indirectly determined the severity of the conditions based on their treatment status. Through our analyses, we were able to show that stroke patients who are being treated with depression have a poorer QoL. This suggests that, compared to patients who have only been diagnosed and are not being treated, patients who are being treated have more severe impairment in QoL. Moreover, we compared the relationships of each of the domains of EQ-5D with the stroke treatment group, depression treatment group, and concurrent stroke and depression treatment group, and the concurrent stroke and depression treatment group had the strongest associations with all of the domains.

16. 349-352, should describe in Materials and Methods section.

- We humbly appreciate the reviewer’s comment and revised the manuscript as below:

Methods

From 45,741 adults included in this study, the participants were first grouped by prevalence of stroke, and further subdivided by prevalence of depression.

Discussion

…we divided the participants into four groups according to disease prevalence of stroke and depression and indirectly determined the severity of the conditions based on their treatment status.

17. Line 358, "strongest association", positive or negative or something should be described here.

- We appreciate the reviewer’s comment and revised the manuscript as below:

concurrent stroke and depression treatment group had the most substantial impairment of all of the domains related to HRQoL

---

## [Decision Letter · Decision Letter 1]

13 May 2022

Association between depression and quality of life in stroke patients: The Korea National Health and Nutrition Examination Survey (KNHANES) IV–VII (2008–2018)

PONE-D-21-40161R1

Dear Dr. Ha,

We’re pleased to inform you that your manuscript has been judged scientifically suitable for publication and will be formally accepted for publication once it meets all outstanding technical requirements.

Kind regards,

Petri Böckerman

Academic Editor

PLOS ONE

Additional Editor Comments (optional):

Reviewers' comments:

Reviewer's Responses to Questions

**Comments to the Author**

1. If the authors have adequately addressed your comments raised in a previous round of review and you feel that this manuscript is now acceptable for publication, you may indicate that here to bypass the “Comments to the Author” section, enter your conflict of interest statement in the “Confidential to Editor” section, and submit your "Accept" recommendation.

Reviewer #1: All comments have been addressed

Reviewer #3: All comments have been addressed

2. Is the manuscript technically sound, and do the data support the conclusions?

Reviewer #1: Yes

Reviewer #3: Yes

3. Has the statistical analysis been performed appropriately and rigorously? 

Reviewer #1: Yes

Reviewer #3: Yes

4. Have the authors made all data underlying the findings in their manuscript fully available?

Reviewer #1: Yes

Reviewer #3: Yes

5. Is the manuscript presented in an intelligible fashion and written in standard English?

Reviewer #1: Yes

Reviewer #3: Yes

6. Review Comments to the Author

Reviewer #1: (No Response)

Reviewer #3: (No Response)

7. PLOS authors have the option to publish the peer review history of their article (what does this mean?). If published, this will include your full peer review and any attached files.

Reviewer #1: No

Reviewer #3: No

---

## [Editor Report · Acceptance letter]

3 Jun 2022

PONE-D-21-40161R1 

Association between depression and quality of life in stroke patients: The Korea National Health and Nutrition Examination Survey (KNHANES) IV–VII (2008–2018) 

Dear Dr. Ha:

I'm pleased to inform you that your manuscript has been deemed suitable for publication in PLOS ONE. Congratulations! Your manuscript is now with our production department. 

Kind regards, 

on behalf of

Professor Petri Böckerman 

Academic Editor

PLOS ONE